# Flows Beat Diffusions on Image Synthesis — and There Are Good Reasons Why

## Abstract

In the DDPM paper, Ho, Jain, and Abbeel introduced two reversible diffusion processes parameterized by a noise schedule—a generator and an oracle process that the generator learns from—and derived a formula for the Kullback-Leibler divergence (KL) in the form of a time-weighted Mean Squared Error (MSE). However, they empirically found that omitting the weights improved performance on image-synthesis benchmarks, a result later corroborated by many studies. More recently, removing the stochastic component at generation time has proved effective. (1) In this work, we provide a theoretical justification for these practices. We consider a broader class of diffusion processes (not necessarily reversible) parameterized by a noise schedule and a diffusion size b that share the same marginals. Since the weight associated with the MSE depends on b, omitting the weight is equivalent to solving the equation weight(b)=1, which yields a unique *MSE-diffusion*. For SOTA models, we checked that b is close to zero; that is, the learned MSE-diffusion is nearly a flow, and we confirm this observation by comparing generators on ImageNet 512×512. Therefore, flows beat reversible diffusions because training of SOTA models is an implementation of KL minimization for MSE-diffusions, which are nearly flows. The models that succeed are the ones that are really trained. (2) Moreover, by generalizing the diffusion process to both discrete and continuous time, we obtained a novel representation of the diffusion state as the sum of an explicit linear component, an unweighted pathwise integral of the denoiser, and a noise term. This representation offers the advantages of DPM-solvers while enabling the use of classical numerical methods for ODEs.

## 1 Introduction

A typical diffusion model for image generation transforms noise into an image over a few dozen to a few hundred time steps by means of a neural network, a neural denoiser. The denoiser is trained with a time-weighted mean-squared error (MSE) between the noised image and the network's prediction. Averaging is performed over independent trios (time, noise, image), thus no diffusion model is needed to train the denoiser. The weights associated with MSE are often absorbed into the noised image, yielding different parameterizations of the prediction (e.g., noise-, data-, or velocity parameterization). Therefore, the denoiser is determined by the noise schedule and the chosen parameterization.

On the other hand, a diffusion model is specified by its noise schedule and diffusion size (variance-rate coefficient). To train a diffusion model, we need a measure of proximity between 2 diffusion processes: a generator and an oracle process that the generator learns from. The processes are defined by distributions, and MSE does not determine the proximity between them, so we may use the most popular measure for this purpose, that is the Kullback-Leibler divergence (KL) or, equivalently, the maximum likelihood (ML).

This brings us to the problem: how to combine the practical training of the denoiser with the theoretical training of the diffusion process, and then determine the diffusion size for a generator based on the noise schedule and parameterization used in training. This issue has been known since at least the work of Ho et al. (2020). They expressed the KL between reversible diffusion processes as a time-weighted MSE. However, empirical results showed that dropping the weights improved performance—a finding later confirmed by many studies. Since then, the problem has become more entrenched because several different noise schedules and parameterizations have appeared, while

diffusion models are defined using a noise schedule and the reversibility condition (Ho et al., 2020; Song et al., 2021c; Nichol & Dhariwal, 2021; Dhariwal & Nichol, 2021; Salimans & Ho, 2021; Song et al., 2021b; Kingma et al., 2021; Ho & Salimans, 2022; Rombach et al., 2022; Kingma & Gao, 2023; Esser et al., 2024). More recently, it has also proved effective to zero out the diffusion size during sampling, replacing the reversible diffusion with a deterministic flow. We validate this claim in the review of state-of-the-art (SOTA) models presented in Section 6.

**Main result.** In this work, we provide a theoretical justification for these practices. We consider a broader class of diffusion processes (not necessarily reversible) parameterized by a noise schedule and a diffusion size that share the same marginals. In Proposition 1 and Proposition 2, we generalize the KL formulas from the Ho et al. (2020) and Kingma et al. (2021) papers, respectively, for this class of models. Subsequently in Proposition 3, we derive a closed-form expression for the diffusion size of the unique *MSE-diffusion* learned by KL minimization, given a noise schedule and a denoiser parameterization. Moreover, in Section 6, we checked that the MSE-diffusion size is close to zero for current SOTA models; that is, the MSE-diffusion is nearly a flow. We then confirm this observation by comparing generators on the ImageNet 512×512 class-conditional benchmark. Therefore, the practical training of denoisers for generators of deterministic flows can be understood as an implementation of KL minimization for MSE-diffusion processes, which are nearly flows. The models that succeed are the ones that are really trained.

**Additional results.** By generalizing the diffusion process to both discrete and continuous time, we can use elementary autoregressive arguments, yielding formulas that are simpler than those used so far. In particular, in Section 3, we obtained a novel representation of the diffusion state as the sum of an explicit linear component, an unweighted pathwise integral of the denoiser, and a noise term. This representation offers the advantages of DPM solvers (Lu et al., 2023; 2022; Cui et al., 2025), while additionally enabling the use of classical methods of numerical integration for differential equations.

## 2 A DENOISER INDUCED BY MSE TRAINING

Assume that we have two positive, continuously differentiable functions of time $t \in (0,1)$, namely increasing *signal schedule* $\alpha_t$ and decreasing *noise schedule* $\sigma_t$. Let $\mathring{\rho}_t = \alpha_t/\sigma_t$ and $\mathring{\lambda}_t = \log \mathring{\rho}_t$ denote *signal-to-noise ratio* and its logarithm, respectively. The ring accents over the symbols indicate that these functions are special cases of more general functions, without rings, which will be defined later. Note also that in our setting, unlike in the original works, a signal schedule is an increasing function. Let $t \sim U(0,1)$, $X \sim p_x$ in $\mathbb{R}^d$ and $\varepsilon \sim \mathcal{N}(0, I_d)$ be independent. We consider the *linear noise generators*

$$\bar{Y}_t := X + \mathring{\rho}_t^{-1}\varepsilon \quad \text{and} \quad \bar{Z}_t := \alpha_t \bar{Y}_t = \alpha_t X + \sigma_t \varepsilon. \tag{1}$$

We train a *prediction* $\hat{u}_t : \mathbb{R}^d \to \mathbb{R}^d$ by fitting its parameters, denoted as a hat, according to the *mean squared error* (MSE)

$$\min_{\wedge} \mathbb{E}_{t,\varepsilon,X} \left\| \hat{u}_t(\alpha_t \bar{Y}_t) - u_t \right\|^2 = \min_{\wedge} \int_0^1 \mathbb{E}_{\varepsilon,X} \left\| \hat{u}_t(\alpha_t \bar{Y}_t) - u_t \right\|^2 \, \mathrm{d}t, \tag{2}$$

where $t \sim U(0,1)$, $u_t = A_t \bar{Y}_t + S_t \varepsilon$ is a target and functions $A_t, S_t$ are scaling schedules, with positive and continuous $S_t$ called *parameterization*. Observe that averaging is performed over independent trios $(t, \varepsilon, X)$, thus no diffusion process is needed to train the denoiser. From a unique $\hat{u}_t$ we recover an estimator of the *noise* from the formula for the target $u_t$ and a *denoiser* or a *data* estimator via (1)

$$\hat{\varepsilon}_t(\bar{Y}_t) := \frac{\hat{u}_t(\alpha_t \bar{Y}_t) - A_t \bar{Y}_t}{S_t} \quad \text{and} \quad \hat{X}_t(\bar{Y}_t) := \bar{Y}_t - \mathring{\rho}_t^{-1}\hat{\varepsilon}_t(\bar{Y}_t). \tag{3}$$

A direct calculation shows that $\hat{u}_t - u_t = S_t(\hat{\varepsilon}_t - \varepsilon) = -\mathring{\rho}_t S_t(\hat{X}_t - X)$. We can also define a target using data: $u_t = B_t \bar{Y}_t - C_t X$, then for this target learn the network, define $\hat{X}_t$, and then, using (1), define $\hat{\varepsilon}_t$ and set $S_t := C_t/\mathring{\rho}_t$.

For our purposes, the interface between the denoiser and the generator consists of, in addition to $\hat{\varepsilon}_t$ or $\hat{X}_t$, the pair $(\mathring{\lambda}_t, S_t)$. These can be viewed as input and output scalings, respectively. Kingma & Gao (2023) showed that MSE-training is determined by $\mathring{\lambda}_t$ and weights $\omega_t = S_t^2/\mathring{\lambda}_t'$ equivalent to our $S_t$.

We now present the most popular noise schedules and parameterizations, including those used in current SOTA models.

**Noise schedules.** Kingma & Gao (2023) demonstrated that three popular noise schedules can be derived uniformly as quantile functions of bell-shaped densities: normal, logistic, and hyperbolic secant.

*Normal.* $\mathring{\lambda}_t := \Phi^{-1}(t) + 0.4$, $\alpha_t := 1$, $\sigma_t = \exp(-\mathring{\lambda}_t)$, where $\Phi$ is the standard normal cumulative distribution function. This schedule is used in the EDM2 model (Karras et al., 2024b;a).

*Logis.* $\alpha_t := t$, $\sigma_t := 1 - t$, $\mathring{\lambda}_t = \log(t/(1-t))$, that is, the quantile of the (standard) logistic distribution. This schedule is inspired by optimal transport between the noise and data distributions (Lipman et al., 2023; Liu et al., 2023; Albergo & Vanden-Eijnden, 2023; Albergo et al., 2023; Ma et al., 2024; Esser et al., 2024; Yao et al., 2025).

*Sech.* $\alpha_t := \sin(\pi t/2)$, $\sigma_t := \cos(\pi t/2)$, $\mathring{\lambda}_t = \log(\tan(\pi t/2))$, so $(2/\pi)\,\mathring{\lambda}_t$ is the quantile function of the hyperbolic secant distribution (Nichol & Dhariwal, 2021; Dhariwal & Nichol, 2021; Kingma et al., 2021).

*SechInter.* Hoogeboom et al. (2023; 2025) proposed the shifted Sech schedule interpolation, which can be expressed in a simplified form as

$$\mathring{\lambda}_t := \log(\tan(\pi t/2)) + (\log 16)(t-1), \ \alpha_t := (1 + \exp(-2\mathring{\lambda}_t))^{-1/2}, \ \sigma_t^2 = 1 - \alpha_t^2.$$

**Parameterizations.**

*Noise.* The network predicts $\varepsilon$, thus $S_t \equiv 1$. This parameterization is undoubtedly the most popular (Ho et al., 2020; Song et al., 2021a;c; Nichol & Dhariwal, 2021; Dhariwal & Nichol, 2021; Rombach et al., 2022; Peebles & Xie, 2023; Hoogeboom et al., 2023; Chen et al., 2024b;a).

*Data.* The network is trained to predict $X$, so $S_t = \mathring{\rho}_t^{-1}$. This natural schedule is mainly of theoretical importance (Kingma et al., 2021; Lu et al., 2022).

*F-pred.* The target for this prediction is $F_t := \sqrt{4 + \mathring{\rho}_t^2}\, X - \mathring{\rho}_t^2/\sqrt{4 + \mathring{\rho}_t^2}\, Y_t$, thus $S_t = \sqrt{4\mathring{\rho}_t^{-2} + 1}$. It is an effective schedule used in the EDM and EDM2 models (Karras et al., 2022; 2024b;a).

*Vel.* The target is the velocity of $\bar{Z}_t$, that is, $v_t := \alpha_t' X + \sigma_t'\varepsilon = \alpha_t'\bar{Y}_t - \mathring{\lambda}_t'\sigma_t\varepsilon$. Then (3) yields $\hat{v}_t(\alpha_t\bar{Y}_t) = \alpha_t'\bar{Y}_t - \mathring{\lambda}_t'\sigma_t\hat{\varepsilon}_t(\bar{Y}_t)$, so $S_t = \mathring{\lambda}_t'\sigma_t$. This parameterization is motivated by considerations from optimal transport theory and is closely related to the logistic noise schedule (Lipman et al., 2023; Liu et al., 2023; Albergo & Vanden-Eijnden, 2023; Albergo et al., 2023; Ma et al., 2024; Yu et al., 2025; Leng et al., 2025; Zheng et al., 2025).

*VelLN.* $S_t := \mathring{\lambda}_t'\sigma_t \exp\!\left(-\frac{1}{4}\left(\log\frac{t}{1-t}\right)^2\right)\!/\!\left((2\pi)^{1/4}\sqrt{t(1-t)}\right)$. This parameterization results from a velocity-type prediction that employs a logit-normal distribution for time sampling (Esser et al., 2024; Yao et al., 2025).

*Sigmoid.* $S_t := \sqrt{2\mathring{\lambda}_t'/\left(1 + \exp(2\mathring{\lambda}_t + 3)\right)}$. This parameterization was introduced in (Kingma & Gao, 2023) and is employed in the SiD2 model (Hoogeboom et al., 2025).

## 3 DIFFUSION MODELS INDUCED BY THE DENOISER

A diffusion denoising model that generates from the distribution estimator $p_x$ is a diffusion process with a drift estimated by a denoiser, starting from pure noise. In this section, we will first define the discrete-time diffusion noise. Then, by adding a denoiser, we will obtain a denoising process, and we will subsequently define an analogous process in continuous time and a numerical solver for it. Finally, we will specify the generation interval and state our main problem.

**Stationary, time-inhomogeneous autoregression.** Let us fix a grid $0 < t_0 < t_1 < \cdots < t_N = t \le t_{max} < 1$, where $t_i = t_0 + i(t_{max} - t_0)/N, i = 0, 1, \ldots, N$. Let $\rho_t$ be a positive, increasing, continuously differentiable function of time $t \in (0, 1)$. This function, which we will refer to as the

*diffusion schedule*, defines the cumulative relative variance of the diffusion processes. We also define $r_t = \rho_t \mathring{\rho}_t$ and $\lambda_t = \log \rho_t$.

Let $\{\xi_{t_i}\}_{i=0}^N$ and $\varepsilon_{t_0} \equiv \varepsilon_0$ be i.i.d. $\mathcal{N}(0, I_d)$ and for $t = t_i, s = t_{i-1}$ set

$$\varepsilon_t := \frac{\rho_s}{\rho_t} \varepsilon_s + \sqrt{1 - \frac{\rho_s^2}{\rho_t^2}} \, \xi_s. \tag{4}$$

The rescaled $\varepsilon_t, \varepsilon_s$ form an autoregressive process with additive noise

$$\rho_t \varepsilon_t = \rho_s \varepsilon_s + \sqrt{\rho_t^2 - \rho_s^2} \, \xi_s. \tag{5}$$

From (5), it is clear that $\varepsilon_t \sim \mathcal{N}(0, I_d)$, and that $\varepsilon_s, \xi_s$ are independent. The correlation and conditional variance are also easily computable for coordinates $j, k = 1, 2, \ldots, d$

$$\mathrm{cor}(\varepsilon_{t,j}, \varepsilon_{s,k}) = \mathbf{1}(j = k)\rho_s/\rho_t, \ \mathbb{V}(\varepsilon_{t,j} \mid \varepsilon_{s,k}) = 1 - \mathbf{1}(j = k)\rho_s^2/\rho_t^2. \tag{6}$$

These formulas indicate that $\rho_t^2$ represents the relative cumulative variance of the process $\{\varepsilon_{t_i}\}_{i=0}^N$.

**Discrete diffusion models.** Assume that $\{\varepsilon_{t_i}\}_{i=0}^N$ are independent of $X$. Let us define

$$Y_t := X + \mathring{\rho}_t^{-1} \varepsilon_t \ \text{ and } \ Z_t := \alpha_t Y_t = \alpha_t X + \sigma_t \varepsilon_t. \tag{7}$$

It is clear that for any diffusion schedule $\rho_t$, the random variables $Y_t, Z_t$ are distributed identically to the linear noise generators $\bar{Y}_t, \bar{Z}_t$ defined in (1) at times $t = t_i$. By substituting the expressions for $\varepsilon_t, \varepsilon_s$ in terms of $Y_t, Y_s$ or $Z_t, Z_s$ (from (7)) into (5), we get

$$r_t Y_t = r_s Y_s + (r_t - r_s) X + \sqrt{\rho_t^2 - \rho_s^2} \, \xi_s. \tag{8}$$

$$\frac{\rho_t}{\sigma_t} Z_t = \frac{\rho_s}{\sigma_s} Z_s + (r_t - r_s) X + \sqrt{\rho_t^2 - \rho_s^2} \, \xi_s. \tag{9}$$

Thus, the function $\alpha_t$, which was previously shown to be an internal learning function, is now independently found to be unnecessary for generation. It is sufficient to generate the process $Y_t$ and, if necessary, scale it at the end of generation to obtain $Z_{t_N} = \alpha_{t_N} Y_{t_N}$. Hence, in this section, we only consider the process $Y_t$. Obviously, the process (8) is not a true generator because it uses $X$, so we will call it an *oracle*.

By substituting the prediction induced by the denoiser, $\hat{X}_s \equiv \hat{X}_s(\hat{Y}_s)$, for $X$ in the oracle process (8), we obtain the *generator*

$$\hat{Y}_{t_0} := \mathring{\rho}_{t_0}^{-1} \varepsilon_{t_0} \ \text{ and } \ r_t \hat{Y}_t := r_s \hat{Y}_s + (r_t - r_s) \hat{X}_s + \sqrt{\rho_t^2 - \rho_s^2} \, \xi_s. \tag{10}$$

Note that in our setting, unlike in many works on generative diffusion models, time in the oracle and the generator runs forward.

**Continuous diffusion models.** By induction, for any grid points $t_i < t_j$ we obtain from (10)

$$r_{t_j} \hat{Y}_{t_j} = r_{t_i} \hat{Y}_{t_i} + (r_{t_{i+1}} - r_{t_i})\hat{X}_{t_i} + \ldots + (r_{t_j} - r_{t_{j-1}})\hat{X}_{t_{j-1}} + \sqrt{\rho_{t_j}^2 - \rho_{t_i}^2} \, \xi_{t_i}^*, \tag{11}$$

where $\hat{Y}_{t_i}, \xi_{t_i}^*$ are independent and

$$\xi_{t_i}^* := \frac{\sqrt{\rho_{t_{i+1}}^2 - \rho_{t_i}^2} \, \xi_{t_i} + \ldots + \sqrt{\rho_{t_j}^2 - \rho_{t_{j-1}}^2} \, \xi_{t_{j-1}}}{\sqrt{\rho_{t_j}^2 - \rho_{t_i}^2}} \sim \mathcal{N}(0, I_d).$$

Assuming $t \mapsto \hat{X}_t(\hat{Y}_t)$ is continuous and $N$ approaches infinity in (11), we arrive at a new representation of the diffusion state as the sum of an explicit linear component, an unweighted pathwise integral of the denoiser, and a noise term

$$r_t \hat{Y}_t = r_s \hat{Y}_s + \int_{r_s}^{r_t} \hat{X}_r(\hat{Y}_r) \, \mathrm{d}r + \sqrt{\rho_t^2 - \rho_s^2} \, \xi_s^*, \tag{12}$$

where $\hat{Y}_s, \xi_s^*$ are independent, $\xi_s^* \sim \mathcal{N}(0, I_d)$ and $t_0 < s < t \leq t_{max} < 1$.

Given (12), we do not need the SDEs. Moreover, the appropriate SDE can be naturally derived from (10); subsequently, using the variation of constants method on the SDE, we can obtain (12) which, with the simplest discretization, takes the form of (10) (the construction is provided in Appendix A).

**Universal diffusion solver (UDS).** Let $t = s + \Delta t$, $q_{st} := r_s/r_t$ and $\beta_{st} := \mathring{\rho}_t^{-1}\sqrt{1 - \rho_s^2/\rho_t^2}$, then from (12) we obtain the solver family

$$\hat{Y}_t := q_{st}\hat{Y}_s + (1 - q_{st})\text{APPROX}\big[\mathbb{E}_{q\sim U(q_{st},1)}\hat{X}_q(\hat{Y}_q)\big] + \beta_{st}\,\xi_s^*, \tag{13}$$

where APPROX denotes any numerical ODE integration method. In the simplest cases, formula (13) yields the Euler and Heun schemes, with the Heun equation depending on a prior calculation of the Euler equation

$$\text{Euler:} \quad \hat{Y}_t := q_{st}\hat{Y}_s + (1 - q_{st})\hat{X}_s(\hat{Y}_s) + \beta_{st}\,\xi_s^*, \tag{14}$$

$$\text{Heun:} \quad \hat{Y}_t := q_{st}\hat{Y}_s + (1 - q_{st})\frac{\hat{X}_s(\hat{Y}_s) + \hat{X}_t(\hat{Y}_t)}{2} + \beta_{st}\,\xi_s^*. \tag{15}$$

**1.** Observe that the deterministic component of the UDS scales with $X$ (usually pre-normalized) as we mix the previous prediction $\hat{Y}_s$ with the mean prediction over the transformed interval $(s, t)$. Hence, the coefficient $\beta_{st}$ of the standard normal noise warrants the name *diffusion size* and can be used for an equivalent definition of the diffusion process. **2.** It is easy to check that EulerUDS is identical to the DDPM solver (Ho et al., 2020) for reversible diffusions, $\beta_{st} = \mathring{\rho}_t^{-1}\sqrt{1 - \mathring{\rho}_s^2/\mathring{\rho}_t^2}$, and to the DDIM solver (Song et al., 2021a) for deterministic flows, $\beta_{st} = 0$. **3.** Universal schemes such as the Euler-Maruyama method struggle with integrating the rapidly-changing linear term, whereas schemes designed specifically for diffusion, such as DPM-solvers, treat the linear term analytically. However, they require integrating $\hat{\varepsilon}_t$ or $\hat{X}_t$ with an exponential weight, a procedure which is complicate (Lu et al., 2023; 2022; Cui et al., 2025). In particular, DPM solvers of order 1-3 are analogous to the Runge-Kutta methods, but the analogue for the Runge-Kutta method of order 4, has not yet been developed. For comparison, our method is both universal and specific to diffusion.

**Generation interval and the main problem.** Comparing the formulas for popular noise schedules with the formula for the generator (10), we see that $\mathring{\rho}_0 = 0$, $\mathring{\rho}_1 = \infty$ and we do not start at the moment when the generator and oracle have the same distribution, nor do we reach the point where the oracle has the distribution $p_x$. To precisely define the generation task, we need to specify the start $t_0$ and end $t_{max}$ of the generation. From formula (10), it is also clear that the function $\lambda_t$ is not needed for generation, only its quotients. Equivalently, it is sufficient to calculate $\lambda_t$ from the integral formula based on the derivative $\lambda_t'$, hereafter referred to as the *diffusion rate*. At this point, we can define the main problem of our work: determine $\lambda_t'$ based on the training of the denoiser $(\mathring{\rho}_t, S_t)$ and the generation interval $0 < t_0 < t_{max} < 1$.

## 4   A DIFFUSION MODEL INDUCED BY THE PENALIZED MAXIMUM LIKELIHOOD

We need a measure of proximity between the oracle process and the generator process to choose the diffusion schedule. The processes are defined by distributions, and MSE does not determine the proximity between them, so we will use the most popular measure for this purpose, that is the ML or, equivalently, the KL.

### 4.1   DIVERGENCE DECOMPOSITIONS

As in the previous sections, we start with the processes $Z_t = \alpha_t Y_t$ and $\hat{Z}_t = \alpha_t \hat{Y}_t$ to see that it is enough to consider only $Y_t$ and $\hat{Y}_t$. Let $t_0 < t_1 < \cdots < t_N = t$ be a discretization of the time interval $[t_0, t]$, set $s = t_{N-1}$. For $x \sim p_x$ we denote latent variables along the path by $z_{t_i}$ and write $z_{t_i:t} = (z_{t_i}, z_{t_{i+1}}, \ldots, z_t)$. From the definition of the oracle process (9) and the generator (10) it follows that

$$p_t(z_t|z_s, x) = \mathcal{N}\Big(z_t|\mu_s(z_s, x), \sigma_t\sqrt{1 - \rho_s^2/\rho_t^2}I_d\Big),$$

$$\text{where} \quad \mu_s(z_s, x) := \frac{\sigma_t\rho_s}{\sigma_s\rho_t}z_s + \frac{\sigma_t}{\rho_t}(\rho_t\mathring{\rho}_t - \rho_s\mathring{\rho}_s)X = \frac{\alpha_t}{\alpha_s}z_s - \sigma_t\Big(\frac{\mathring{\rho}_t}{\mathring{\rho}_s} - \frac{\rho_s}{\rho_t}\Big)\varepsilon_s.$$

Analogously

$$\hat{p}_t(z_t|z_s) = \mathcal{N}\Big(z_t|\hat{\mu}_s(z_s), \sigma_t\sqrt{1 - \rho_s^2/\rho_t^2}I_d\Big),$$

$$\text{where} \quad \hat{\mu}_s(z_s) := \frac{\sigma_t\rho_s}{\sigma_s\rho_t}z_s + \frac{\sigma_t}{\rho_t}(\rho_t\mathring{\rho}_t - \rho_s\mathring{\rho}_s)\hat{X}_s = \frac{\alpha_t}{\alpha_s}z_s - \sigma_t\Big(\frac{\mathring{\rho}_t}{\mathring{\rho}_s} - \frac{\rho_s}{\rho_t}\Big)\hat{\varepsilon}_s.$$

The KL divergence between these two normal distributions is

$$\mathbb{D}[p_t(.|z_s,x) \mid \hat{p}_t(.|z_s)] := \mathbb{E}_{Z_t|z_s,x} \log\left[p_t(Z_t|z_s,x)/\hat{p}_t(Z_t|z_s)\right] \tag{16}$$

$$= \frac{\|\mu_s(z_s,x) - \hat{\mu}_s(z_s)\|^2}{2\sigma_t^2(1-\rho_s^2/\rho_t^2)} = w_s^N \|\hat{\varepsilon}_s(z_s) - \varepsilon_s\|^2, \text{ where } w_s^N := \frac{(\mathring{\rho}_t/\mathring{\rho}_s - \rho_s/\rho_t)^2}{2(1-\rho_s^2/\rho_t^2)}. \tag{17}$$

We define the following conditional and joint distributions

$$p_{t_1:t}^N(z_{t_1:t}|z_{t_0},x) := p_t(z_t|z_s,x)\dots p_{t_1}(z_{t_1}|z_{t_0},x),$$

$$\hat{p}_{t_1:t}^N(z_{t_1:t}|z_{t_0}) := \hat{p}_t(z_t|z_s)\dots\hat{p}_{t_1}(z_{t_1}|z_{t_0}),$$

$$p_{t_0,x}(z_{t_0},x) := p_{t_0}(z_{t_0}|x)\,p_x(x),$$

$$\overline{p}_{t_0,t}(z_{t_0},x|z_t) := p_{t_0}(z_{t_0})\,\overline{p}_t(x|z_t),$$

$$p_{t_0:t,x}^N(z_{t_0:t},x) := p_{t_1:t}^N(z_{t_1:t}|z_{t_0},x)p_{t_0,x}(z_{t_0},x),$$

$$\hat{p}_{t_0:t,x}^N(z_{t_0:t},x) := \hat{p}_{t_1:t}^N(z_{t_1:t}|z_{t_0})\overline{p}_{t_0,t}(z_{t_0},x|z_t).$$

The distribution $\overline{p}_t$ represents the "reconstruction error," which determines how well the image was recovered from the final $z_t$ representation. The overline symbol indicates its parameters, and its effect on the divergence between the joint distributions will be called the *bias* and denoted below as $\bar{\mathcal{B}}$. Two KL decompositions that we shall use are

$$\mathbb{D}\left[p_{t_0:t,x}^N \,\middle\|\, \hat{p}_{t_0:t,x}^N\right] = \mathbb{D}\left[p_x \,\middle\|\, \hat{p}_x^N\right] + \mathbb{E}_X\,\mathbb{D}\left[p_{t_0:t}^N(.|X) \,\middle\|\, \hat{p}_{t_0:t}^N(.|X)\right], \tag{18}$$

$$\mathbb{D}\left[p_{t_0:t,x}^N \,\middle\|\, \hat{p}_{t_0:t,x}^N\right] = \hat{\mathcal{L}}^N(t_0,t,\mathring{\lambda},\lambda) + \bar{\mathcal{B}}(t_0,t), \tag{19}$$

where

$$\hat{\mathcal{L}}^N(t_0,t,\mathring{\lambda},\lambda) := \mathbb{E}_{Z_{t_0},X}\,\mathbb{D}\left[p_{t_1:t}^N(.|Z_{t_0},X) \,\middle\|\, \hat{p}_{t_1:t}^N(.|Z_{t_0})\right] \text{ and } \bar{\mathcal{B}}(t_0,t) := \mathbb{D}\left[p_{t_0,x} \,\middle\|\, \overline{p}_{t_0,t}\right]. \tag{20}$$

Both of these decompositions together imply that the *diffusion loss*, denoted as $\hat{\mathcal{L}}^N$, is the objective function for (implicitly) a penalized negative log-likelihood of the estimator $\hat{p}_x^N$ of the density $p_x$ induced by the denoiser with parameters $\wedge$.

**Proposition 1.**

$$\hat{\mathcal{L}}^N(t_0,t,\mathring{\lambda},\lambda) = \sum_{i=0}^{N-1} w_{t_i}^N \mathbb{E}_\varepsilon \|\hat{\varepsilon}_{t_i} - \varepsilon\|^2. \tag{21}$$

**Proposition 2.** *Assuming $t \mapsto \hat{\varepsilon}_t(\hat{Y}_t)$ is continuous, we have*

$$\hat{\mathcal{L}}^N(t_0,t,\mathring{\lambda},\lambda) = \hat{\mathcal{L}}(t_0,t,\mathring{\lambda},\lambda') + \mathcal{O}(1/N), \tag{22}$$

*where*

$$\hat{\mathcal{L}}(t_0,t,\mathring{\lambda}',\lambda') := \int_{t_0}^t \frac{\left(\lambda_\tau' + \mathring{\lambda}_\tau'\right)^2}{4\,\lambda_\tau'} \mathbb{E}_{\varepsilon,X} \|\hat{\varepsilon}_\tau(Y_\tau) - \varepsilon\|^2 \, d\tau. \tag{23}$$

Propositions 1-2 are proven in Appendix A.

## 4.2 PENALIZED MAXIMUM LIKELIHOOD

Observe that the weights under the integral in (23) are of the form

$$\frac{\left(\lambda_t' + \mathring{\lambda}_t'\right)^2}{4\,\lambda_t'} = \mathring{\lambda}_t' + \frac{1}{4}\chi^2(\lambda_t', \mathring{\lambda}_t'), \text{ where } \chi^2(\lambda_t', \mathring{\lambda}_t') := \frac{\left(\lambda_t' - \mathring{\lambda}_t'\right)^2}{\lambda_t'} \tag{24}$$

is the well-known $\chi^2$-distance. So weights determining the diffusion process in $\hat{\mathcal{L}}$ are, up to a constant, scaled distance between $\lambda_t'$ and $\mathring{\lambda}_t'$.

Since the function $\hat{\mathcal{L}}$ is (implicitly) a penalized ML objective, we do not change its meaning or difficulty of its calculation, if we add a simple penalty to the weights and define (explicitly) the *penalized maximum likelihood* objective

$$\hat{\mathcal{L}}_c(t_0,t,\mathring{\lambda},\lambda') := \int_{t_0}^t \left[\mathring{\lambda}_\tau' + \frac{\chi^2(\lambda_t',\mathring{\lambda}_t') + c_t\lambda_t'}{4}\right]\mathbb{E}_{\varepsilon,X} \|\hat{\varepsilon}_\tau(Y_\tau) - \varepsilon\|^2 \, d\tau, \tag{25}$$

for some non-negative, continuous function $c_t$. Indeed, it is easy to check that the optimal diffusion rate for such penalized weights is

$$\lambda'_{c,t} := \mathring{\lambda}'_t / \sqrt{1 + c_t}. \tag{26}$$

Penalized maximum likelihood covers many important approaches. If $c_t = 0$, then we obtain the maximum (joint) likelihood solution, hereinafter denoted as *ML-diffusion*, which is, as previously shown (Ho et al., 2020; Kingma et al., 2021; Kingma & Gao, 2023), a reversible diffusion. If $c_t \to \infty$, then $\lambda'_{c,t} \to 0$, and a diffusion process converges to a deterministic flow. Our MSE-diffusion, defined in the next section, is a tradeoff between these extremes.

## 5 A DIFFUSION MODEL INDUCED BY MSE TRAINING

We want the losses to agree not only globally on the interval $[t_0, t_{max}]$, but also on each of its subintervals. Let us imagine a scenario where the group of researchers optimizing the reconstruction error improves its method and thus decreases $t_{max}$, or when it becomes possible to start the generation process for a larger $t_0$. It could also be that we should generate diffusions in stages using different samplers, and our sampler might only care about optimality for a certain subinterval. Below we will formulate an appropriate condition, but first let us define MSE for each initial interval $(t_0, t)$

$$\hat{\mathcal{M}}(t_0, t, \mathring{\lambda}, S) := \int_{t_0}^t \mathbb{E}_{\varepsilon,X} S_t^2 \left\| \hat{\varepsilon}_t(\bar{Y}_t) - \varepsilon_t \right\|^2 \, \mathrm{d}t = \int_{t_0}^t \mathbb{E}_{\varepsilon,X} \left\| \hat{u}_t(\alpha_t \bar{Y}_t) - u_t \right\|^2 \, \mathrm{d}t. \tag{27}$$

We will say that the diffusion process defined by $(t_0, t, \mathring{\lambda}', \lambda'_c)$ is *coherent with MSE* if and only if the following condition is satisfied

**Coherence principle.** There exist a constant $M \equiv M(t_0, t_{max}, \mathring{\lambda}, S)$ such that $\forall t \in [t_0, t_{max}]$ we have

$$\hat{\mathcal{L}}(t_0, t, \mathring{\lambda}', \lambda'_c) = M \hat{\mathcal{M}}(t_0, t, \mathring{\lambda}, S). \tag{28}$$

The loss $\hat{\mathcal{L}}$ (without subscript c) is invariant to data scaling, because it is the expected divergence, whereas MSE depends on data scaling. Therefore, to compare the two functions, we need an appropriate normalization, that is some constant $M$.

**Proposition 3.** *Let us define*

$$M := \max_{t \in [t_0, t_{max}]} \mathring{\lambda}'_t / S_t^2. \tag{29}$$

*Then the coherence principle holds with M iff the diffusion rate is*

$$\lambda'_{t,c} = \left( \sqrt{M} S_t - \sqrt{M S_t^2 - \mathring{\lambda}'_t} \right)^2. \tag{30}$$

The diffusion process with a parameter $\check{\lambda}'_t \equiv \lambda'_{t,c}$ is called the *MSE-diffusion*. Proposition 3 is proven in Appendix A.

**Example.** Consider the logistic noise schedule with the velocity parametrization: $\alpha_t := t, \sigma_t := 1 - t, \mathring{\lambda}'_t = 1/[t(1-t)], S_t = 1/t$ and $g_t := \sqrt{(t_{max} - t)/(t_{max}(1-t))}$.

Thus $M = t_{max}/(1 - t_{max})$ and

$$\check{\lambda}'_t = \left( \sqrt{M} - \sqrt{M - t/(1-t)} \right)^2 \Big/ t^2.$$

In this case, we obtain a compact form for $\check{\lambda}_t$

$$\check{\lambda}_t = -\log\left(\frac{t}{1-t}\right) - \log\left(\frac{1+g_t}{1-g_t}\right) + \frac{2\,t_{max}}{(1-t_{max})}\frac{1-t}{t}\left(g_t - 1\right) + const.$$

**Discrete time.** The coherence principle and MSE-diffusion have their equivalent in the discrete model. Let us define $\eta_{st} := \sqrt{1 - \rho_s^2/\rho_t^2} = \mathring{\rho}_t \beta_{st}$ and $s := t - \Delta t$, $\Delta t := (t_{max} - t_0)/N$. Rewriting equation (4) we see that $\eta_{st}^2$ is the proportion of new noise $\xi_s$ to the total noise $\varepsilon_t$

$$\varepsilon_t = \sqrt{1 - \eta_{st}^2}\,\varepsilon_s + \eta_{st}\,\xi_s.$$

From (17)

$$w_t^N(\eta_{st}) = \frac{1}{\eta_{st}^2}\left(\sqrt{\gamma_{st}} - \sqrt{1 - \eta_{st}^2}\right)^2, \text{ where } \gamma_{st} := \mathring{\rho}_t^2/\mathring{\rho}_s^2.$$

It can be easily checked that

$$\mathring{\eta}_{st} := \arg\min_{\eta} w_t^2(\eta) = \sqrt{1 - \gamma_{st}^{-1}},$$

and

$$\check{\eta}_{st} = \frac{\gamma_{st} - 1}{\sqrt{2MS_t^2\gamma_{st}} + \sqrt{2MS_t^2 + 1 - \gamma_{st}}}, \text{ where } M := \max_{t \in \{t_0, \ldots, t_{max}\}} (\gamma_{st} - 1)/(2S_t^2).$$

To calculate the diffusion size $\beta_{t-\Delta t, t}$ in the next section, we employ both $\lambda_t'$ and $\eta_{t-\Delta t, t}$ using the approximation $2\Delta t \lambda_t' \approx \eta_{t-\Delta t, t}^2$.

# 6 EXPERIMENT AND DISCUSSION

In this section, we present a concise review of SOTA models and argue, based on experimental evidence, that their corresponding MSE-diffusions are almost flows. We also briefly discuss additional results. Let $ML(\mathring{\rho}_t)$ and $MSE(\mathring{\rho}_t, S_t)$ denote the diffusion size $\beta_{t-\Delta t, t}$ for ML-diffusion and MSE-diffusion, respectively.

**Current open-source SOTA "diffusion models" generate deterministic flows.** It is natural to group SOTA "diffusion models" for image generation according to their neural network architecture—either DiT (Peebles & Xie, 2023) or U-Net (Ronneberger et al., 2015). This applies to both ImageNet class-conditioned models and text-to-image models.

The largest group consists of DiT-based models, which can be conveniently divided into two subgroups depending on whether Saining Xie is listed as a co-author. The subsequent models by Xie include DiT (Peebles & Xie, 2023), SiT (Ma et al., 2024), REPA (Yu et al., 2025), REPA-E (Leng et al., 2025) and RAE (Zheng et al., 2025). The original DiT employed the noise schedule, parameterization, and generator from DDPM, whereas the later models use the *Logis* noise schedule and *Vel* parameterization. It is instructive to examine how the generators evolved across these models: initially, a reversible diffusion $ML(Logis)$ generator from DDPM was used (DiT); then a limited diffusion $REPA(Logis)$ generator defined by $\lambda_t' = 1/(2t)$ was adopted (SiT, REPA, REPA-E); and finally, a deterministic flow $\beta_{t-\Delta t, t} = 0 \approx MSE(Logis, Vel)$ emerged (RAE)—see Figure 1. This evolution of Xie's models—from reversible diffusion to flows—provides empirical confirmation of our claim that SOTA models fit MSE-diffusions. Other DiT-based models include LightningDiT (Yao et al., 2025) and popular text-to-image models such as SD3 (Esser et al., 2024) and PixArt (Chen et al., 2024b;a). All of them generate flows. The SD3 and LightningDiT models use the *Logis* noise schedule and *VelLN* parameterization resulting from the use of importance sampling in velocity parameterization, while PixArt uses a beta-linear noise schedule and *Noise* parameterization. Thus, the latest SOTA "diffusion models" based on DiT generate deterministic flows.

SOTA "diffusion models" based on U-Net originate from the DDPM network and its mature modification, ADM (Dhariwal & Nichol, 2021), which operated with the *Sech* noise schedule and *Noise* parameterization and used the DDPM generator. These models can be further divided according to the research group. The first subgroup consists of models developed at NVIDIA: EDM (Karras et al., 2022) and EDM2 (Karras et al., 2024b;a). These models use a *Normal* noise schedule, $F - pred$ parameterization, and generate flows. The second subgroup consists of models developed at Google: SiD (Hoogeboom et al., 2023) and SiD2 (Hoogeboom et al., 2025). These models use a *SechInter* noise schedule and noise and sigmoid parameterization, respectively. They are the only SOTA "diffusion models" known to us that generate reversible diffusion (approximately, due to interpolation between the prior and posterior variance), but they are not publicly available.

**MSE-diffusions are nearly flows.** In Figure 1, we show the diffusion size $\beta_{t-\Delta t,t}$ for ML-diffusion and MSE-diffusion for the current SOTA models. It can be observed that the MSE-diffusion is close to zero for all models except at the initial stage of generation, which should not matter given that the generator starts from a normal distribution and the diffusion noise is also normal. In the later stages of generation, the diffusion size is below $1\%$ in all models, with the exception of EDM2, where it remains at $5\%$ of the image size. Therefore, we decided to verify whether such values imply that the flow generator and the MSE-diffusion are practically indistinguishable in terms of FID in image generation. We conducted an experiment comparing generators in the EDM2 environment using version S with CFG, applying all settings including the selection of time steps, specific to this model. The results are shown in Table 1. As expected, the MSE-diffusion performs nearly like a flow, whereas HeunUDS (15) in the flow version performs worse than the original HeunEDM2—likely due to the choice of the EDM2 non-uniform time grid.

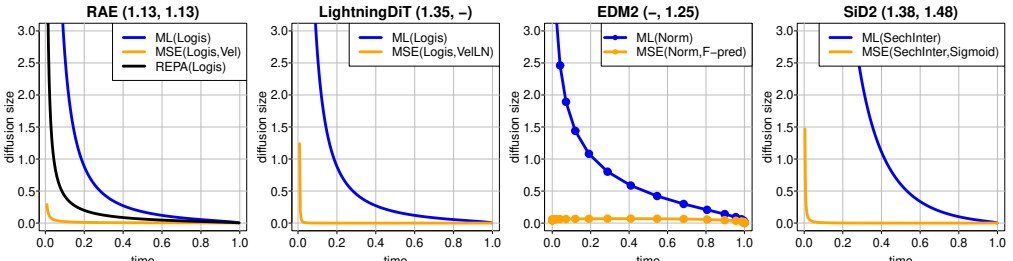

Figure 1: **Diffusion sizes of current SOTA models.** The FID scores for the ImageNet $256{\times}256$ and ImageNet $512{\times}512$ class-conditional benchmarks are indicated in parentheses next to the model names. In the case of EDM2, the generator uses only 32 steps on a non-uniform grid, which we have marked with dots. In all other cases, the generators operate on a uniform grid, with default settings of 250 steps for RAE and LightningDiT, and 512 steps for SiD2.

| NFE | HeunEDM2 Flow | HeunUDS, eq. (15) | | | EulerUDS, eq. (14) | | |
|---|---|---|---|---|---|---|---|
| | | Flow | MSE | ML | Flow | MSE | ML |
| 63 | 2.28 | 2.45 | 2.45 | 3.39 | 2.81 | 2.84 | 4.30 |
| 255 | 2.26 | 2.27 | 2.28 | 2.71 | 2.34 | 2.36 | 2.84 |

Table 1: **The FID scores for the ImageNet $512{\times}512$ in the EDM2 environment.** We run the generators in the environment of the EDM2 version S with CFG, leaving all settings unchanged. NFE denotes the number of function evaluations. We compute FID 5 times in each experiment and report the mean.

**Remarks. 1.** We generalize the generative diffusion model and the formulas for the diffusion loss from Kingma et al. (2021) and Song et al. (2021b). Formula (23) is equivalent to the 'KLUB' expression introduced in Sabour et al. (2024), but we do not use stochastic calculus in its proof. Despite the generalization, our continuous-time diffusion construction is much simpler than previous ones: time runs forward and there is no need to consider SDEs at all. **2.** Implementing a diffusion model using $\alpha_t$ and $\sigma_t$ has become common practice, despite the mostly simulation-based arguments of Karras et al. (2022), that $\alpha_t$ is unnecessary. We specify these arguments: $\alpha_t$ is merely an input scaling in the denoiser, which is not needed for generation or in the context of maximum likelihood analysis. Our research indicates that the natural scale for the process values is $\lambda_t + \mathring{\lambda}_t$. **3.** We see no significant difference between score-based models that generate processes on $(0, \infty)$ and stochastic interpolants that work on $[0, 1]$. It is important that $\mathring{\rho}_t$ and $\rho_t$ take on positive values within the closed interval of actual generation. This is necessary to make the analysis realistic, which is clearly visible in the proofs of global convergence for numerical ODE solvers (Lu et al., 2023; 2022).

**Conclusion.** SOTA models learn MSE-diffusions; MSE-diffusions are nearly flows; flow generators beat reversible diffusions. The models that succeed are the ones that are really trained.

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

## A  APPENDIX

**From the generator** (10) **to SDE and back.**  The equation (10) for the simplest generator is equivalent to

$$\hat{Y}_t - \hat{Y}_s = \left(\frac{r_s - r_t}{\Delta t\, r_t}\hat{Y}_s + \frac{r_t - r_s}{\Delta t\, r_t}\hat{X}_s\right)\Delta t + \mathring{\rho}_t^{-1}\sqrt{\frac{\rho_t^2 - \rho_s^2}{\Delta t\, \rho_t^2}}\sqrt{\Delta t}\,\xi_s^*. \tag{31}$$

A direct manipulation with Taylor expansions yields, for $s = t - \Delta t$,

$$\frac{r_t - r_s}{\Delta t\, r_t} = (\log r_t)'(1 + \delta_1)\ \text{ and }\ \frac{\rho_t - \rho_s}{\Delta t\, \rho_t}\frac{\rho_t + \rho_s}{\rho_t} = 2(\log \rho_t)'(1 + \delta_2), \tag{32}$$

where $|\delta_1|, |\delta_2| = \mathcal{O}(1/N)$. Hence (31) takes the form

$$\hat{Y}_t - \hat{Y}_s = (\log r_t)'\left(\hat{X}_t - \hat{Y}_t\right)\Delta t + \mathring{\rho}_t^{-1}\sqrt{2(\log \rho_t)'}\sqrt{\Delta t}\,\xi_s^* + \mathcal{O}_P(\Delta t). \tag{33}$$

Let $W_t$ be a standard $d$-dimensional Wiener process and assume that $\hat{X}_t$ is sufficiently regular. Then the difference equation (33) converges to the Itô SDE

$$d\hat{Y}_t = (\log r_t)'\left(\hat{X}_t - \hat{Y}_t\right)dt + \mathring{\rho}_t^{-1}\sqrt{2(\log \rho_t)'}\,dW_t, \tag{34}$$

$$= (\lambda_t' + \mathring{\lambda}_t')\left(\hat{X}_t - \hat{Y}_t\right)dt + \mathring{\rho}_t^{-1}\sqrt{2\lambda_t'}\,dW_t, \tag{35}$$

$$= \mathring{\rho}_t^{-1}(\lambda_t' + \mathring{\lambda}_t')\,\hat{\varepsilon}_t\,dt + \mathring{\rho}_t^{-1}\sqrt{2\lambda_t'}\,dW_t. \tag{36}$$

By substituting $\tilde{Y}_t = (r_t/r_s)\hat{Y}_t$ for $\hat{Y}_t$ in (34) we get

$$r_s d\tilde{Y}_t = r_t'\hat{X}_t\,dt + \rho_t\sqrt{2(\log \rho_t)'}\,dW_t. \tag{37}$$

By integrating (37) and returning to $\hat{Y}_t$, we obtain (12), which, with the simplest discretization, takes the form of (10). $\qquad\square$

Note that the substitution leading to formula (37) uses the method of variation of constants—the same approach is used to derive DPM solvers. Thanks to the construction of the diffusion process by (12), SDEs are not needed.

**Proof of Proposition 1.**  By the chain rule for KL along the grid $t_0 < t_1 < \cdots < t_N = t$ we obtain

$$\mathbb{D}\left[p_{t_1:t}^N(.|z_{t_0}, x)\,\big\|\,\hat{p}_{t_1:t}^N(.|z_{t_0})\right]$$
$$= \mathbb{D}\left[p_{t_1}(.|z_{t_0}, x)\,\big\|\,\hat{p}_{t_1}(.|z_{t_0})\right] + \mathbb{E}_{Z_{t_1}}\mathbb{D}\left[p_{t_2}(.|Z_{t_1}, x)\,\big\|\,\hat{p}_{t_2}(.|Z_{t_1})\right] + \cdots$$
$$+ \mathbb{E}_{Z_{t_{N-1}}}\mathbb{D}\left[p_t(.|Z_{t_{N-1}}, x)\,\big\|\,\hat{p}_t(.|Z_{t_{N-1}})\right]. \tag{38}$$

In our setting, from (17) this can be rewritten in terms of denoising errors with weights $w_{t_j}^N$:

$$(38) = w_{t_0}^N\,\|\hat{\varepsilon}_{t_0} - \varepsilon_{t_0}\|^2 + w_{t_1}^N\,\mathbb{E}_{\varepsilon_{t_1}}\|\hat{\varepsilon}_{t_1} - \varepsilon_{t_1}\|^2 + \cdots + w_{t_{N-1}}^N\,\mathbb{E}_{\varepsilon_{t_{N-1}}}\|\hat{\varepsilon}_{t_{N-1}} - \varepsilon_{t_{N-1}}\|^2. \tag{39}$$

So

$$\mathbb{E}_{Z_{t_0}, X}\mathbb{D}\left[p_{t_1:t}^N(.|Z_{t_0}, X)\,\big\|\,\hat{p}_{t_1:t}^N(.|Z_{t_0})\right] = \sum_{i=0}^{N-1}\mathbb{E}_{\varepsilon_{t_i}, X}w_{t_i}^N\,\|\hat{\varepsilon}_{t_i} - \varepsilon_{t_i}\|^2 = \sum_{i=0}^{N-1}\mathbb{E}_{\varepsilon, X}w_{t_i}^N\,\|\hat{\varepsilon}_{t_i} - \varepsilon\|^2.$$
$$\square$$

**Proof of Proposition 2.**  A direct manipulation with Taylor expansions yields, for $s = t - \Delta\tau$,

$$\bar{w}_{t-\Delta\tau}^N := \frac{w_{t-\Delta\tau}^N}{\Delta\tau} = \left(\frac{\mathring{\rho}_t - \mathring{\rho}_s}{\Delta\tau\,\mathring{\rho}_s} + \frac{\rho_t - \rho_s}{\Delta\tau\,\rho_t}\right)^2 \Big/ \left(2\frac{\rho_t - \rho_s}{\Delta\tau\,\rho_t}\frac{\rho_t + \rho_s}{\rho_t}\right)$$

$$= \left(\mathring{\lambda}_t'(1 + \delta_1) + \lambda_t'(1 + \delta_2)\right)^2 \big/ \left(4\lambda_t'(1 + \delta_3)\right)$$

$$= \bar{w}_t + \mathcal{O}(1/N), \text{ where } \bar{w}_t := \frac{\left(\lambda_t' + \mathring{\lambda}_t'\right)^2}{4\lambda_t'} \text{ and } |\delta_1|, |\delta_2|, |\delta_3| = \mathcal{O}(1/N). \tag{40}$$

For $\tau \in [t_0, t]$ define $t^N(\tau) := \min\{t_i : \tau \geq t_i\}$. We have

$$\max_{t_0 \leq \tau \leq t} \left( \bar{w}^N_{t^N(\tau)} \left\| \hat{\varepsilon}_{t^N(\tau)} - \varepsilon \right\|^2 - \bar{w}_\tau \left\| \hat{\varepsilon}_\tau - \varepsilon \right\|^2 \right) = \mathcal{O}(1/N), \tag{41}$$

and consequently

$$\hat{\mathcal{L}}^N(t_0, t, \mathring{\lambda}, \lambda) = \mathbb{E}_{\varepsilon, X} \left( \sum_{i=0}^{N-1} \frac{w^N_{t_i}}{\Delta\tau} \left\| \hat{\varepsilon}_{t_i} - \varepsilon \right\|^2 \Delta\tau \right)$$

$$= \mathbb{E}_{\varepsilon, X} \left( \int_{t_0}^t \bar{w}^N_{t^N(\tau)} \left\| \hat{\varepsilon}_{t^N(\tau)} - \varepsilon \right\|^2 d\tau \right) = \int_{t_0}^t \bar{w}_\tau \, \mathbb{E}_{\varepsilon, X} \left\| \hat{\varepsilon}_\tau - \varepsilon \right\|^2 d\tau + \mathcal{O}(1/N). \quad \square$$

**Proof of Proposition 3.** Let us fix $t$ and simplify notation $\beta := \lambda'_t, \beta_c := \lambda'_{c,t}, \mathring{\beta} := \mathring{\lambda}'_t, s := \sqrt{M}S_t$. The coherence condition implies that the integrals $\hat{\mathcal{L}}$ and $M\hat{\mathcal{M}}$ agree on the initial intervals, which is equivalent to the equality of the integrands. Therefore

$$\frac{(\beta + \mathring{\beta})^2}{4\beta} = MS_t^2 = s^2. \tag{42}$$

The definition of the constant $M$ implies that $s^2 \geq \mathring{\beta}$, thus equation (42) has 2 roots

$$\beta_- = \left( s - \sqrt{s^2 - \mathring{\beta}} \right)^2 \quad \text{and} \quad \beta_+ = \left( s + \sqrt{s^2 - \mathring{\beta}} \right)^2. \tag{43}$$

Observe that $\beta_- \beta_+ = \mathring{\beta}^2$, so $\beta_- \leq \mathring{\beta} \leq \beta_+$. By comparison with (26) $\beta_- = \beta_{c_-}, \beta_+ = \beta_{c_+}$, we obtain formulas for the penalty constants $c_- = (\mathring{\beta}/\beta_-)^2 - 1$, $c_+ = (\mathring{\beta}/\beta_+)^2 - 1$ and $c_+ \leq 0 \leq c_-$. Hence only $\beta_-$ optimizes the penalized maximum likelihood objective. Sufficiency is obvious. $\quad \square$

