# OpenReview forum: "A Diffusion Model Induced by MSE Training"
_ICLR.cc/2026/Conference — Submitted to ICLR 2026_

### Official Review · Reviewer_3Cp4 · 2025-10-30

**Soundness:** 2
**Presentation:** 2
**Contribution:** 2
**Rating:** 4
**Confidence:** 3

**Summary:**

This paper addresses a common discrepancy in diffusion models where the training process (defined by an MSE objective) differs from the generation process.

The authors propose a "principle of coherence" to align them, deriving a closed-form expression for the generative diffusion schedule based on the specific noise schedule and network parameterization used during training. This "MSE-induced diffusion" is intended to be the process the denoiser actually learned , and the paper also provides a new state representation to simplify numerical integration.

**Strengths:**

1. The paper has a clear motivation, it highlights a common mismatch in diffusion models where the denoiser is trained using one implicit process defined by an MSE objective, noise schedule, and network parameterization but is then used for generation with a different process, e.g., one with an ML-optimal diffusion schedule or a deterministic flow

2. The work derives a closed-form expression for the diffusion schedule. This means the generative process isn't an arbitrary choice but is analytically determined by the noise schedule and network parameterization used during training. This is called "MSE-induced diffusion"

**Weaknesses:**

The most important issue is that the article lacks practical evidence

1. The paper is entirely theoretical. It proposes the MSE-induced diffusion process based on a coherence principle , but it presents no experiments to demonstrate that this new process is stable, effective, or produces better results than the "incoherent" methods it critiques

2. The paper claims its new state representation e.g. "makes it straightforward to apply classical numerical integration methods" and clarifies the relation to DPM-solvers. It even alludes to the potential for new solvers like a 4th-order Runge-Kutta. However, it does not actually implement or test any such solver, so these practical benefits remain hypothetical.

3. The primary evidence for the theory's validity is Figure 1. This figure merely shows a visual similarity between the shape of derived schedule and schedules found empirically in other recent work, e.g., empPML, Discount. This correlation is interesting but does not prove that the coherence principle is the reason for that shape or that the resulting process is effective.

4. The paper critiques widely used deterministic flows as being incoherent. It then proposes its stochastic coherent process as a better alternative. However, it doesn't sufficiently justify why coherence is inherently superior to the fast and high-performing deterministic flows, largely assuming that aligning the training and sampling processes is axiomatically beneficial

**Questions:**

I hope the author can provide enough evidence to prove the effectiveness of the methods and theories. Especially the followings:

1. The core claim is that **it is beneficial to generate samples using the very process that is actually learned**. This is a compelling, testable hypothesis. However, the paper is entirely theoretical, and its primary evidence (Figure 1) is correlational .
Could the authors provide any preliminary empirical results (e.g., FID scores, sample quality on a standard benchmark like CIFAR-10) to support this central hypothesis?

2. The paper makes strong, practical claims about its new state representation (Eq. 12 ), suggesting it **makes it straightforward to apply classical numerical integration methods** and potentially enables new solvers, like an RK4 analogue, which are currently difficult for DPM-solvers. Have the authors performed any proof-of-concept implementations of a numerical solver using this new unweighted pathwise integral formulation?

3. The paper critiques widely used deterministic flows where $\lambda_t' \rightarrow 0$ and explicitly contrasts them with its proposed MSE-induced process, which is stochastic. Given the demonstrated speed and high performance of incoherent deterministic samplers, what is the specific benefit of adhering to the coherence principle?

---

> ### Author Response · Authors · 2025-12-03
>
> **Response to Area Chair**
>
> We are grateful to the reviewers for their valuable feedback. The reviewers collectively raised the same core criticism:
>
> - **88xM**: *Unconvincing Motivation (…) Lack of Experiments*
> - **Y5wa**: *extremely light empirical glimpse*
> - **3Cp4**: *article lack practicle evidence*
>
> We agree with the reviewers. We were initially unsure what the implications of our results were for image generation with diffusion models. However, we have spent over a month studying the relevant literature, and thanks to the reviewers' feedback, we have fundamentally improved the paper.
>
> We are submitting a new version of the paper. While it contains no new theoretical results or computationally expensive experiments, the changes are fundamental. The paper is no longer primarily about *MSE-diffusion*, but offers a theoretical explanation for why, in practice, deterministic flows beat reversible diffusions. *MSE-diffusion* now serves as the means to answer this question.
>
> ---
>
> ### A. Main Result
>
> 1. In Section 5, we prove that the practical training of SOTA models, i.e., MSE-training, is equivalent to KL-divergence minimization for our *MSE-diffusion*.
> 2. In the new Section 6, we show, based on calculations and experiments, that *MSE-diffusions* are nearly flows.
> 3. Section 6 also presents a concise review of SOTA models from which the following conclusion clearly arises:
>    *“Current open-source SOTA ‘diffusion models’ generate deterministic flows”*, that is, flows beat reversible diffusions.
>
> **Conclusion.** SOTA models learn *MSE-diffusions*; *MSE-diffusions* are nearly flows; flow generators beat reversible diffusions. The models that succeed are the ones that are effectively trained.
>
> ---
>
> ### B. Remaining Results
>
> 1. In the current version of the paper, we implemented the Universal Diffusion Solver (UDS) within the EDM2 environment. Using UDS, we obtained our main result that *MSE-diffusions* are nearly flows. We showed a proof-of-concept implementation, although our HeunUDS in the flow version performs worse (by about 7%) than the original HeunEDM2—likely due to the choice of the EDM2 non-uniform time grid. Improving the performance of UDS is a topic for future work.
> 2. The key difference from the previous version is the $\beta$ coefficient, referred to as the *diffusion size*. This coefficient is the working characteristic of the generator and shows how diffusion differs from a flow (definition: line 216, interpretation: lines 226–231). It is thanks to the diffusion size that we understood that *MSE-diffusion* is nearly a flow.
>
> ---
>
> ### C. Responses to Questions
>
> The reviewers did not question the theoretical results contained in Sections 2–5, and these have not undergone significant changes. Nevertheless, we have added a new Section 6, *“Experiment and Discussion”*, changed the title, and completely rewritten the Abstract and Introduction.
>
> #### Reviewer 88xM
>
> - **Q1–Q2.** The Coherence Principle and Proposition 3 provide a formula for the diffusion that minimizes MSE. This allows us to link practical training (MSE training, in which diffusion does not appear explicitly) with KL minimization, i.e., fitting a diffusion.
>
> #### Reviewer 3Cp4
>
> - **Q1–Q2.** We performed the experiments suggested by the reviewer.
> - **Q3.** In the previous version, we did not explicitly criticize deterministic flows, but rather promoted *MSE-diffusion*. It has now become clear that *MSE-diffusion* is nearly a flow, allowing us to explain the superiority of flows over reversible diffusion.
> - We are pleased that the reviewer, who understood the significance of our work better than we initially did and suggested crucial changes, provided us with a positive rating.

---

### Official Review · Reviewer_Y5wa · 2025-10-30

**Soundness:** 1
**Presentation:** 2
**Contribution:** 1
**Rating:** 0
**Confidence:** 4

**Summary:**

The paper proposes a principle of coherence between the training and generation process of the diffusion model: if a denoiser is trained with time-weighted MSE under a given noise schedule and parameterization, then the generation process should use a matching diffusion schedule derived in closed form in the paper. The authors analyze both discrete- and continuous-time settings via simple autoregressive arguments and introduce a state representation that makes connection to classic ODE solvers straightforward.

**Strengths:**

The paper derives an analytical formula for the diffusion schedule given the training noise schedule and network parameterization.

**Weaknesses:**

The paper is predominantly analytical and offers only extremely light empirical glimpse—mainly a figure comparing shapes of diffusion schedules under various noise/parameterization choices. The idea behind analysis is not deep enough, the analysis itself is not mathematically challenging. Although the work argues its state representation makes classic ODE solvers like RK4 straightforward to use, it does not demonstrate numerical benefits versus modern diffusion solvers. Overall the paper lacks a clear result and falls well below the standard of ICLR.

**Questions:**

None

---

### Official Review · Reviewer_88xM · 2025-11-05

**Soundness:** 1
**Presentation:** 2
**Contribution:** 2
**Rating:** 2
**Confidence:** 3

**Summary:**

This paper proposes a framework called “MSE-Induced Coherent Diffusion.” Starting from weighted MSE training (given a noise schedule and parameterization), the authors introduce an “MSE–ML Coherence Principle,” which provides a closed-form expression for the generative diffusion rate (Proposition 3). This leads to a generative process theoretically consistent with the training objective. They also reformulate the diffusion state as a combination of a linear term, an unweighted path integral, and a noise term, enabling direct use of standard ODE solvers (e.g., RK4). The paper argues that in practice, researchers often “train one schedule but sample with another” (e.g., ML-optimal or deterministic zero-diffusion flows), and advocates using a generative process coherent with the training objective, providing closed-form and discrete formulations consistent with recent empirical practices.

**Strengths:**

**Theoretical Novelty – The Coherence Principle.**

A major ambiguity in diffusion models lies in the disconnect between training and inference processes. The authors attempt to resolve this by introducing the Coherence Principle, which asserts that the empirical MSE loss and the theoretical ML objective should be proportional over any time interval $[t_0, t]$. This is a novel and well-defined (strong) theoretical assumption.

**Weaknesses:**

1. Unconvincing Motivation.

The authors claim to eliminate inconsistency between training and inference noise schedules, but this is unnecessary. The core idea of diffusion models (e.g., DDPM, VP-SDE, Rectified Flow) is distribution matching between the forward and reverse processes. As long as the marginal distributions match, the generative model is valid. There are infinitely many possible paths that share the same marginal distributions—for example, infinitely many SDEs corresponding to the same VP-SDE marginals, or an ODE with equivalent marginals. Once the reverse parameter (score, noise, or velocity) is learned, there theoretically exist infinitely many valid sampling schemes. Therefore, enforcing the Coherence Principle is not inherently necessary.

2. Lack of Experiments / Related Work.

The authors argue that the noise schedule determined by the Coherence Principle is beneficial, citing a few works (Cui et al., 2025; Ma et al., 2024) that allegedly conform to it. However, this reasoning is flawed. The examples are too limited, as many SOTA methods do not satisfy the Coherence Principle, yet were selectively omitted. Moreover, no experiments are presented. If deterministic sampling is said to violate the principle, the authors must show that all schedules coherent with it outperform deterministic ODE sampling; otherwise, the claimed advantage of the Coherence Principle remains unsubstantiated.

**Questions:**

1. Can the authors justify the necessity of the Coherence Principle? It is a very strong assumption, yet the paper devotes too little space to explaining or motivating it.
2. Can the validity of the Coherence Principle be verified through extensive ablation and reasoning experiments, rather than being limited to the few cited works?

---

### Meta-Review · Area_Chair_BF2x · 2025-12-15

**Summary:**

This paper proposes a coherence principle and an MSE-Induced Diffusion framework to theoretically align the training objective with the generative diffusion process. All reviewers recommended rejection, citing a significant lack of empirical validation. The primary criticism is that while the theoretical derivation of a closed-form diffusion schedule is mathematically sound, the paper failed to demonstrate that this alignment is practically necessary or beneficial compared to existing state-of-the-art methods (e.g., deterministic flows). The reviewers found the motivation unconvincing, noting that marginal distribution matching, standard in current diffusion models, does not require the strict coherence proposed here. Additionally, claims regarding improved numerical solvers were initially hypothetical and unsupported by data.

**Reviewer Concerns:**

**Addressed:** The authors acknowledged the lack of experimental evidence and, during the rebuttal, attempted to address this by implementing a *Universal Diffusion Solver* within the EDM2 framework. Authors also significantly rewrote the Abstract and Introduction to pivot the paper’s narrative toward explaining why deterministic flows outperform reversible diffusions.

**Outstanding:** Despite the pivot, the core concern regarding the utility and performance of the proposed method remains outstanding.

  - Performance: The authors admitted in their rebuttal that their proof-of-concept implementation performed worse (approximately 7% degradation) than the standard HeunEDM2 baseline. Consequently, the claim that the proposed method is effective remains unproven.

  - Motivation: The reviewers questioned the necessity of the "Coherence Principle." The authors' rebuttal attempts to reframe the paper as an explanatory theory for why flows work, but this represents a fundamental shift in the paper's identity late in the review process, essentially acknowledging that the original framing of "MSE-Induced Diffusion" as a superior generative method was not supported.

  - Numerical Solvers: The claims regarding the advantages of the new state representation for ODE solvers remain theoretical without demonstrated numerical benefits over existing solvers.

**Reviewer Scores:**

- Reviewer 88xM: 2 (My guess, reviewer would have kept their score). The rebuttal did not provide the compelling performance evidence requested to justify the strong theoretical assumptions.

  - Reviewer Y5wa: 0 or 2 (My guess, reviewer would have kept their score or increase it to 1, 2 but still remain on the very negative side). The added experiment showed inferior performance, reinforcing the initial assessment that the work lacks practical contribution.

  - Reviewer 3Cp4: 4 (My guess, reviewer would have kept their score). While the authors provided the requested experiment, the negative result confirms the reviewer's skepticism about the effectiveness of the method.

---

### Decision · Program_Chairs · 2026-01-26

Reject